# Heparan sulphate binding controls in vivo half-life of the HpARI protein family

Florent Colomb[1], Abhishek Jamwal[2,3], Adefunke Ogunkanbi[1,4], Tania Frangova[1], Alice R Savage[1], Sarah Kelly[1], Gavin J Wright[4], Matthew K Higgins[2,3], Henry J McSorley[1]*

[1]Division of Cell Signalling and Immunology, School of Life Sciences, University of Dundee, Dundee, United Kingdom; [2]Department of Biochemistry, University of Oxford, Oxford, United Kingdom; [3]Kavli Institute of Nanoscience Discovery, Dorothy-Crowfoot Hodgkin Building, University of Oxford, Oxford, United Kingdom; [4]Department of Biology, Hull York Medical School, York Biomedical Research Institute, University of York, York, United Kingdom

## eLife Assessment

This **important** study uses in vitro and in vivo methods to identify HpARI proteins from H. polygyrus as modulators of the host immune system. The data from comprehensive approaches for investigating differential roles of HpARI proteins are **convincing**. This paper is relevant to those who investigate host-pathogen interactions at the systems and molecular levels.

**\*For correspondence:**
hmcsorley001@dundee.ac.uk

**Competing interest:** The authors declare that no competing interests exist.

**Abstract** The parasitic nematode *Heligmosomoides polygyrus bakeri* secretes the HpARI family, which bind to IL-33, either suppressing (HpARI1 and HpARI2) or enhancing (HpARI3) responses to the cytokine. We previously showed that HpARI2 also bound to DNA via its first complement control protein (CCP1) domain. Here, we find that HpARI1 can also bind DNA, while HpARI3 cannot. Through the production of HpARI2/HpARI3 CCP1 domain-swapped chimeras, DNA-binding ability can be transferred, and correlates with in vivo half-life of administered proteins. We found that HpARI1 and HpARI2 (but not HpARI3) also binds to the extracellular matrix component heparan sulphate (HS), and structural modelling showed a basic charged patch in the CCP1 domain of HpARI1 and HpARI2 (but not HpARI3) which could facilitate these interactions. Finally, a mutant of HpARI2 was produced which lacked DNA and HS binding, and was also shown to have a short half-life in vivo. Therefore, we propose that during infection the suppressive HpARI1 and HpARI2 proteins have long-lasting effects at the site of deposition due to DNA and/or extracellular matrix interactions, while HpARI3 has a shorter half-life due to a lack of these interactions.

## Introduction

IL-33 is a pleiotropic cytokine. It can act as a potent inducer of type 2 immune responses, and genetic studies show a strong link between the IL-33 pathway and the risk of developing allergic asthma (*Moffatt et al., 2010*; *Oboki et al., 2010*). Likewise, IL-33 responses are important for induction of anti-parasite type 2 immune responses in helminth infections (*McSorley and Smyth, 2021*). Conversely, IL-33 is a potent inducer of IFN-γ and regulatory T cell responses, dependent on the cytokine's context, cytokine release mode, and the cell responding to the cytokine (*Bonilla et al., 2012*; *Schiering et al., 2014*). Thus, IL-33 can have pro-allergic, pro-inflammatory, or anti-inflammatory activities, and modulation of IL-33 responses can have a range of effects.

Parasitic helminths secrete an array of immunomodulatory proteins which tailor the host immune response to allow parasite persistence without immunopathology (*Maizels et al., 2018*). Our previous work identified the HpARI protein family, secreted by the murine intestinal nematode *Heligmosomoides polygyrus bakeri*. Each member of the HpARI family consists of three complement control protein domains, CCP1-3 (*Colomb et al., 2024*). Our initial characterisation of the HpARI protein (later renamed HpARI2) showed that it bound directly to IL-33, inhibiting the interaction between IL-33 and its receptor (*Osbourn et al., 2017*). Our recent structural characterisation of the HpARI2:IL-33 complex showed that HpARI2 binds IL-33 via interactions between the cytokine and the CCP2 and CCP3 domains of HpARI2. In particular, a long loop in CCP3 contacts IL-33 at a site which overlaps with the IL-33:ST2 binding site, thus inhibiting cytokine-receptor interactions to effectively block the cytokine (*Jamwal et al., 2023*). However, the CCP1 domain was not characterised in this structure.

We previously showed that HpARI2 can interact with DNA in a non-sequence-specific manner, via its CCP1 domain. Truncations of HpARI2 which lack CCP1 cannot bind to DNA (*Osbourn et al., 2017*), while HpARI2 truncations lacking the CCP3 domain can still bind to IL-33 (albeit with lower affinity), but cannot block interactions between IL-33 and its receptor (*Chauché et al., 2020*; *Jamwal et al., 2023*). Thus, through CCP1-DNA and CCP2/3-IL-33 interactions, HpARI2 binds to genomic DNA and IL-33 within the nucleus of necrotic epithelial cells, tethering the cytokine and preventing its release (*Osbourn et al., 2017*). Characterisation of the other members of the HpARI family showed that HpARI1, like HpARI2, blocked responses to IL-33. HpARI3, however, bound to IL-33 but did not block interaction of IL-33 with its receptor. Instead HpARI3 stabilised IL-33, amplifying responses to the cytokine in vivo (*Colomb et al., 2024*). Thus, *H. polygyrus bakeri* produces a family of structurally similar HpARI molecules with opposing physiological effects.

Here, we investigated the role of the CCP1 domain of the HpARI proteins, and found that HpARI1 and HpARI2 can also bind to heparan sulphate (HS) via their CCP1 domain, while HpARI3 cannot. This extracellular matrix binding extends the half-life of HpARI2 in vivo at the site of administration.

## Results
### HpARI1 and HpARI2 bind to DNA, while HpARI3 does not

HpARI2 was previously shown to tether IL-33 in necrotic cell nuclei (*Osbourn et al., 2017*). Using a short-term in vivo model of IL-33 release, we co-administered HpARI1, HpARI2, or HpARI3 with Alternaria allergen (*Figure 1A*) and assessed IL-33 release. As shown previously (*Osbourn et al., 2017*), due to interaction between the HpARIs and IL-33, released IL-33 could not be detected by enzyme-linked immunosorbent assay (ELISA) in the presence of HpARIs (*Figure 1B*). However, denaturing western blots dissociated the HpARI-IL-33 complex and allowed measurement of released IL-33 (*Figure 1C* and *Figure 1—figure supplement 1*). As shown previously, HpARI2 suppressed the release of IL-33 measured by western blot, due to its tethering function (*Osbourn et al., 2017*). HpARI1 showed a similar trend which did not reach statistical significance in this experiment. HpARI3 lacked this ability and did not alter the levels of released IL-33 as assessed by western blot. We therefore measured the ability of the HpARIs to bind to DNA (as a correlate of the tethering function of these proteins) using an electromobility shift assay (EMSA) to measure formation of DNA-protein complexes via altered polyacrylamide gel migration of short DNA oligonucleotides. HpARI1 and HpARI2 both caused altered migration of DNA probes, indicating binding to DNA, while HpARI3 did not (*Figure 1D*). Interestingly, while HpARI2 effectively bound all DNA, HpARI1 showed a reduced DNA-binding ability, only binding 50% of the DNA in this assay (*Figure 1E*).

The DNA-binding activity of HpARI2 was previously shown to be localised to its CCP1 domain. To assess the stark difference in DNA binding between HpARI2 and HpARI3, fusion constructs were expressed with the CCP1 domain of HpARI2 fused to the CCP2+3 domains of HpARI3 (HpARI2:3) or vice versa (*Figure 1F*). In EMSA experiments, these constructs showed DNA binding directly related to the source of their CCP1 domain, with HpARI3:2 lacking, and HpARI2:3 showing DNA binding (*Figure 1G*). Therefore, members of the HpARI family differ in their ability to bind both IL-33 and DNA, the latter of which is mediated solely by the CCP1 domain.

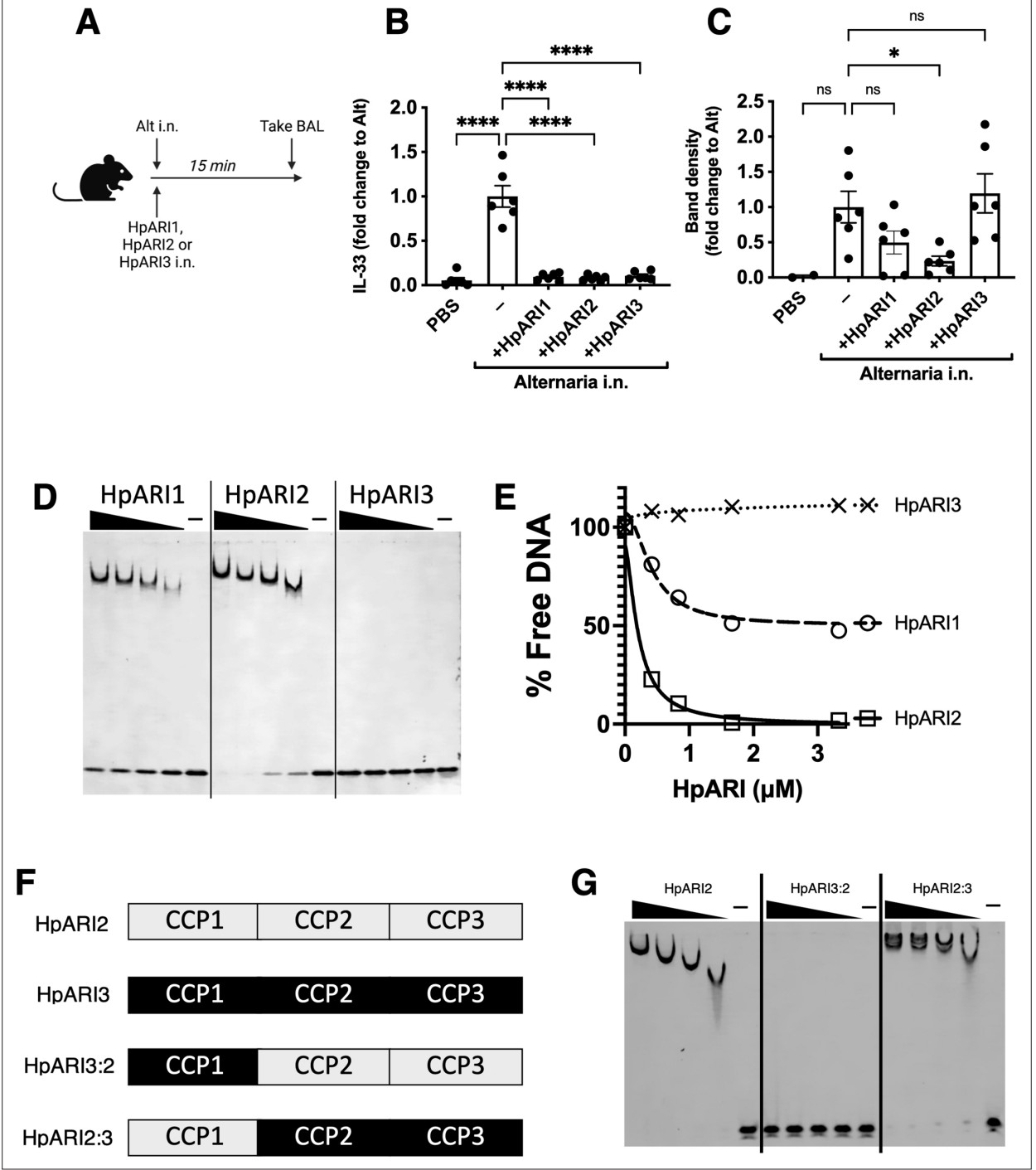

**Figure 1.** HpARI family members have differing effects against IL-33 release. (**A**) Experimental setup for **B** and **C**. (**B**) Quantification of mouse IL-33 levels by enzyme-linked immunosorbent assay (ELISA) in the cell-free bronchoalveolar lavage (BAL) fluids of mice treated as shown in A. (**C**) Quantification of IL-33 by western blot of the samples shown in B. (**D**) Electromobility shift assay (EMSA) of HpARI1, HpARI2, and HpARI3. (**E**) Quantification of % free DNA (i.e. low molecular weight band) in the presence of each protein in EMSA shown in D. (**F**) HpARI3:2 and HpARI2:3 fusion protein design. (**G**) EMSA of HpARI2, HpARI3:HpARI2 fusion, or HpARI2:HpARI3 fusion. Data in B–C pooled from two repeat experiments each with three mice per group for a total n=6. Error bar shows SEM. ns = not significant, *=p<0.05, ****=p<0.0001.

The online version of this article includes the following source data and figure supplement(s) for figure 1:

**Source data 1.** Original western blot and gel images from *Figure 1D and G* and *Figure 1—figure supplement 1*.

**Source data 2.** Annotated western blot and gel images from *Figure 1D and G* and *Figure 1—figure supplement 1*.

**Figure supplement 1.** IL-33 western blot in bronchoalveolar lavage (BAL) supernatant.

## HpARI fusion proteins show unexpected effects on IL-33 responses in vivo

The HpARI2:3 and HpARI3:2 fusion proteins represent useful tools to separate the DNA binding and IL-33 modulation effects of HpARI2 and HpARI3, respectively. These fusion proteins were tested in an in vitro model of IL-33 release and responsiveness (*Chauché et al., 2020*; *Colomb et al., 2024*), where the CMT-64 epithelial cell line is freeze-thawed to induce necrosis and IL-33 release in the presence of HpARI proteins. Supernatants from these necrotic cells were then transferred to cultures of bone marrow cells in the presence of IL-2 and IL-7 to support ILC2 differentiation, and IL-33-mediated ILC2 activation was assessed by IL-5 release. As shown previously (*Colomb et al., 2024*), in this assay HpARI2 dose-dependently suppresses ILC2 responses, while HpARI3 amplifies them (*Figure 2A*). The HpARI2:3 and HpARI3:2 fusion proteins retained the amplifying or suppressing activities of their CCP2/3 domains. Small alterations in activity were seen when CCP1 domains were swapped, with HpARI2:3 showing decreased activity compared to HpARI3 wild-type (HpARI3 EC50=1.2 ng/ml, HpARI3:2 EC50=21.5 ng/ml), and HpARI3:2 showing slightly increased activity compared to HpARI2 wild-type (HpARI2 IC50=1.9 ng/ml, HpARI3:2 IC50=1.1 ng/ml).

As these fusion proteins retained similar activity on IL-33 as their CCP2/3 parent proteins, we then assessed their activity in vivo, first co-administering them with Alternaria allergen and assessing IL-33 release by western blot 15 min later. As expected, HpARI2 and HpARI2:3 tethered IL-33 and reduced its release, due to their combination of IL-33 and DNA-binding activities. HpARI3 and HpARI3:2 lack tethering ability, and slightly increased levels of IL-33 in BAL, possibly due to binding and stabilisation of the soluble cytokine in the absence of DNA binding (*Figure 2B*). Unexpectedly, when the downstream response to released IL-33 was measured 24 hr later (*Figure 2C*), IL-33 tethering did not correlate with suppression of IL-33-induced responses: while HpARI2 suppressed and HpARI3 amplified eosinophilia, ILC2 activation (as measured by surface CD25 or cell size by median FSC), and IL-5 release (*Figure 2D–H*), HpARI2:3 could not suppress IL-33 responses despite tethering of IL-33 at the 15 min timepoint (*Figure 2B*). In fact, HpARI2:3 amplified IL-33-dependent responses similarly to HpARI3, and tended to have increased activity compared to the HpARI3 wild-type protein. Similarly, despite a lack of IL-33 tethering, HpARI3:2 was capable of effectively blocking IL-33-dependent responses similarly to HpARI2 wild-type protein (*Figure 2D–H*). Therefore, these data indicated that our model of DNA tethering of IL-33 by HpARI2 could not fully explain the activity of these proteins.

We hypothesised that the altered activity of the fusion proteins could be due to their effective half-life in vivo. We administered HpARI2 or HpARI3:2 either 1 day, 3 days, or 7 days prior to Alternaria (*Figure 3A*), and found that while HpARI2 could suppress Alternaria-induced eosinophilia and ILC2 responses when administered 3 days prior to allergen, HpARI3:2 could only replicate this suppression when administered 24 hr prior to Alternaria (*Figure 3B–E* and *Figure 3—figure supplements 1–3*). Therefore, the in vivo half-life of HpARI2 is controlled by the CCP1 domain, and the HpARI3 CCP1 domain confers a shorter in vivo half-life than the HpARI2 counterpart. As extracellular DNA is only present in the bronchoalveolar lavage (BAL) at very low concentrations at homeostasis (*Curren et al., 2023*), and is released in response to stimuli such as Alternaria, we hypothesised that the HpARI2 CCP1 could have a second binding partner which mediates its long in vivo half-life. As other host and pathogen CCP domain proteins have been demonstrated to interact with extracellular matrix constituents such as HS (*Mark et al., 2006*; *Schmidt et al., 2008*), we hypothesised that this could be the case for the HpARIs also.

## HpARI2 binds to HS

We first used gel filtration chromatography to assess the mobility of HpARIs in the presence or absence of HS. While HS addition resulted in earlier elution of HpARI1 and HpARI2, no shift was seen with HpARI3, implying a lack of HS binding (*Figure 4A*). Likewise, HpARI1 and HpARI2 could be pulled down with heparin-coated beads while HpARI3 could not (*Figure 4B*). As well as being a common constituent of the extracellular matrix, HS is present on cell surface proteoglycans (*Xu and Esko, 2014*). To assess whether HS binding could mediate interactions between the HpARIs and stromal cells, we added PE-conjugated tetramers of the HpARIs to preparations of naïve mouse lung cells, showing that HpARI1 and HpARI2 bind strongly to CD45-negative stromal cells, but less to CD45-positive cells, while HpARI3 did not show detectable binding to any lung cells (*Figure 4C*). Finally, to assess the respective affinities for HS, we used isothermal calorimetry to show that HpARI1 had

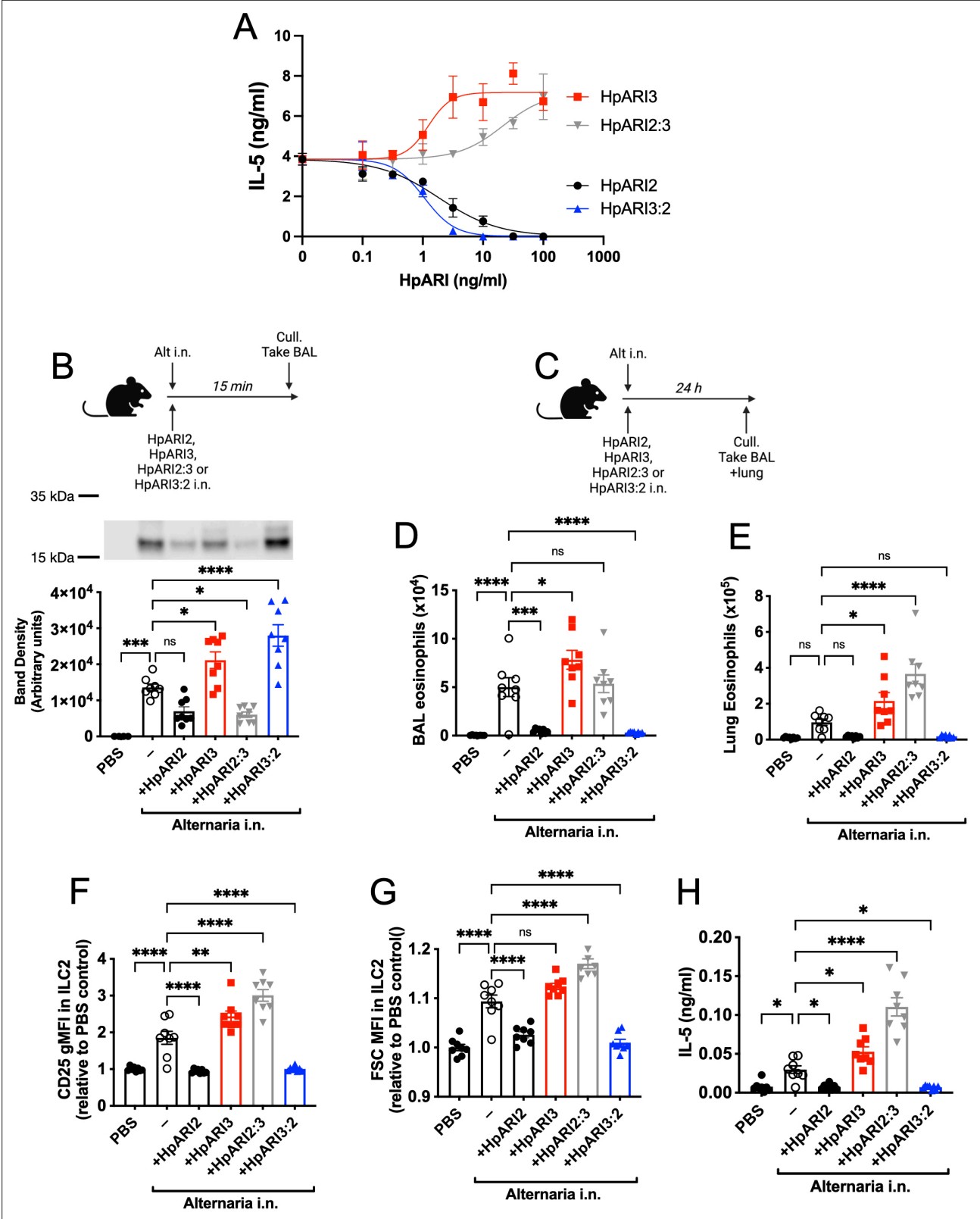

**Figure 2.** HpARI2:HpARI3 chimeras indicate CCP2/3 domains are central to IL-33 amplification versus suppression. (**A**) IL-5 production from bone marrow cells in response to IL-2, IL-7, and freeze-thawed CMT-64 supernatants, in the presence of a range of concentrations of HpARI2, HpARI3, HpARI2:3, or HpARI3:2. Data pooled from three biological replicates. (**B**) Alternaria allergen with HpARI2, HpARI3, or fusions were administered to mice, cell-free bronchoalveolar lavage (BAL) fluid prepared 15 min later, and IL-33 measured by western blot. (**C**) Experimental setup for D–H.

*Figure 2 continued on next page*

*Figure 2 continued*

(**D**) BAL eosinophil numbers (Siglecf⁺CD11⁻CD45⁺) from mice treated as shown in C. (**E**) Eosinophil (Siglecf^hiCD11⁻CD45⁺) numbers in lung tissue from mice treated as shown in C. (**F**) CD25 expression level on lung ILC2 (ICOS⁺Lin⁻CD45⁺) from mice treated as shown in C. (**G**) FSC mean in lung ILC2 (ICOS⁺Lin⁻CD45⁺) from mice treated as shown in C. (**H**) BAL IL-5 levels (enzyme-linked immunosorbent assay [ELISA]) from mice treated as shown in C. All in vivo data pooled from two repeat experiments each with four mice per group for a total n=8. Error bar shows SEM. NS = not significant, *=p<0.05 **=p<0.01, ***=p<0.001, ****=p<0.0001.

The online version of this article includes the following source data for figure 2:

**Source data 1.** Original western blot images from *Figure 2B*.

**Source data 2.** Annotated western blot image from *Figure 2B*.

an affinity for HS of 9.6 µM, while HpARI2 had a slightly higher affinity of 4.2 µM. Furthermore, these experiments confirmed that no binding could be detected between HpARI3 and HS (*Figure 4D–F*).

We next used protein modelling to understand the differential HS binding of different HpARIs. AlphaFold2 models were produced to predict the structures of the CCP1 domains of HpARI1, HpARI2, and HpARI3. This revealed that while HpARI1 and HpARI2 contain a strongly positively charged surface, HpARI3 lacks this charged surface (*Figure 5A*). We next used molecular docking to predict the structure of the complex between HpARI1 or HpARI2 CCP1 domains and the extracellular matrix component HS. Docking experiments predicted that HS molecules can dock to various locations across the positively charged surfaces of HpARI1 and HpARI2 CCP1 domains (*Figure 5B and*

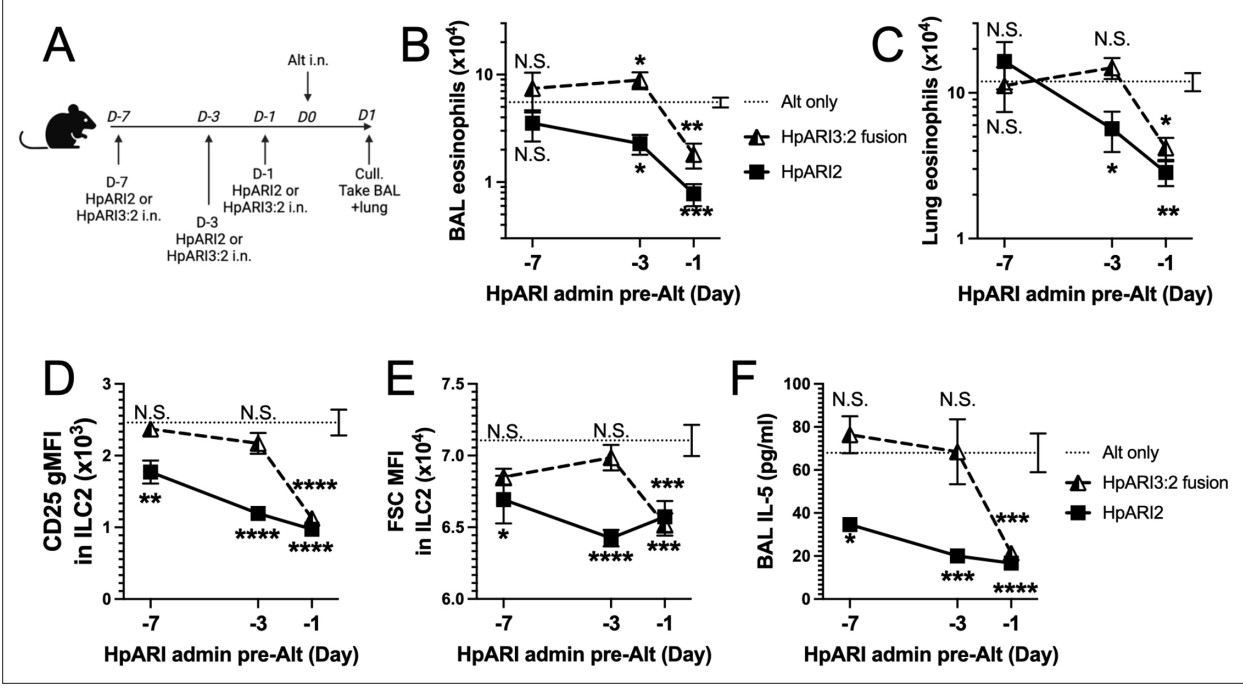

**Figure 3.** HpARI fusion proteins show CCP1 domain determines half-life in vivo. (**A**) Experimental setup for B–E. HpARI2 or HpARI3:2 fusion were administered intranasally 7 days, 3 days, or 1 day prior to Alternaria allergen. Mice were culled 24 hr after Alternaria administration, and bronchoalveolar lavage (BAL) and lung tissue taken for analysis. (**B**) BAL eosinophil numbers (Siglecf⁺CD11⁻CD45⁺) from mice treated as shown in A. (**C**) Eosinophil (Siglecf^hiCD11⁻CD45⁺) numbers in lung tissue from mice treated as shown in A. (**D**) CD25 geometric mean fluorescence intensity on lung ILC2 (ICOS⁺CD90⁺Lin⁻CD45⁺) from mice treated as shown in A. (**E**) FSC mean in lung ILC2 from mice treated as shown in A. (**F**) BAL IL-5 levels (enzyme linked immunosorbent assay [ELISA]) from mice treated as shown in A. Data from day –7 timepoint from a single experimental repeat, all other groups pooled from two experiments. Total biological repeats at day 7=4, all other timepoints n=8. Error bar shows SEM. NS = not significant, *=p<0.05 **=p<0.01, ***=p<0.001, ****=p<0.0001. Analysed by two-way analysis of variance (ANOVA) with Dunnett's post-test.

The online version of this article includes the following figure supplement(s) for figure 3:

**Figure supplement 1.** Gating strategy and representative flow plots for bronchoalveolar lavage (BAL) eosinophils.

**Figure supplement 2.** Representative flow plots for lung tissue cells.

**Figure supplement 3.** Gating strategy and representative flow plots for lung tissue type 2 innate lymphoid cells (ILC2).

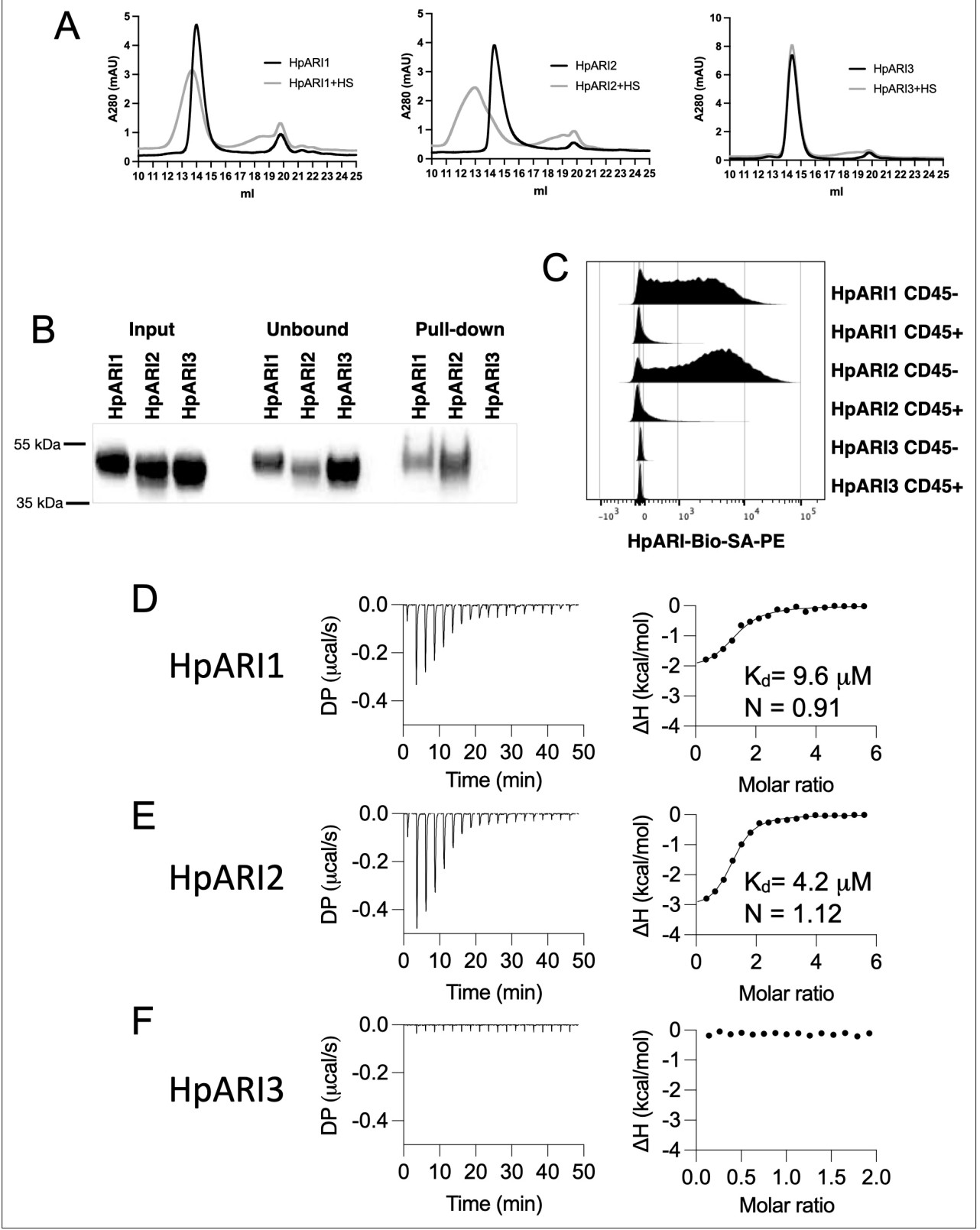

**Figure 4.** HpARI proteins have variable levels of heparin sulphate (HS) binding. (**A**) 50 μg of HpARI1, HpARI2, or HpARI3 were added to 50 μg HS and ran on a Superdex 200 Increase 10/300 GL gel filtration column. A280 trace shown. (**B**) Coomassie gel of HpARI1, HpARI2, or HpARI3 pull-down using HS-coated beads. Input, unbound (i.e. supernatant from beads) and pull-down elution shown. Representative of three repeat experiments. (**C**) Flow cytometry staining of HpARI1, HpARI2, or HpARI3 tetramers with streptavidin-PE, on naive mouse lung cells gated on live CD45+ or live CD45- cells.

*Figure 4 continued on next page*

*Figure 4 continued*

Representatives of two repeat experiments. (**D**) Isothermal calorimetry of HpARI1 ± HS. (**E**) Isothermal calorimetry of HpARI2 ± HS. (**F**) Isothermal calorimetry of HpARI3 ± HS.

The online version of this article includes the following source data for figure 4:

**Source data 1.** Original Coomassie gel images from *Figure 4B*.

**Source data 2.** Annotated Coomassie gel image from *Figure 4B*.

---

*C*). Therefore, the CCP1 domain of HpARI1 and -2 has a positively charged surface which acts as a non-specific binding site for negatively charged cellular components, including DNA and HS, but this binding surface is lacking in HpARI3.

## HpARI2:HS interactions are mediated by CCP1 arginine residues

Electrostatic modelling of HpARI2 indicated that five arginine residues in CCP1 form the positively charged HS- and DNA-binding surface (*Figure 5A*). A pentaR mutant of HpARI2 (with each of the indicated Arg resides mutated to Ala) was expressed. HpARI2_pentaR had identical IL-33 suppressive activity in vitro as compared to HpARI2 wild-type (*Figure 6A*), indicating its IL-33 binding was unaffected. As predicted, however, HpARI2_pentaR lacked binding to both heparin (*Figure 6B*) and DNA (*Figure 6C*). When HpARI2_pentaR was administered to mice prior to Alternaria administration, like HpARI3:2, it was found to have a shorter in vivo half-life than the wild-type HpARI2 protein (*Figure 6D–H*). Taken together, this data shows that HpARI2 is an HS-binding protein, and that interactions with HS increase its in vivo half-life at the site of administration.

## Discussion

The parasite helminth *H. polygyrus bakeri* expresses a range of immunomodulatory proteins which have distinct activities. For instance, the TGF-β mimic Hp-TGM binds to the mammalian TGF-β receptor as well as cell surface co-receptors (*Johnston et al., 2017*; *van Dinther et al., 2023*). There is a family of 10 Hp-TGM proteins (Hp-TGM1-10) (*Smyth et al., 2018*) which have distinct activities: HpTGM-1 is an agonist of the TGF-β receptor and uses CD44 as a co-receptor, directing its activity to CD44-positive immune cells such as CD4[+] T cells, resulting in regulatory T cell induction (*van Dinther et al., 2023*). Conversely, in a recent pre-print the related protein Hp-TGM6 was shown to be a TGF-β receptor antagonist which is particularly active on fibroblasts (*White et al., 2023*), blocking TGF-β signalling. Thus the prototypic Hp-TGM sequence has undergone selection to carry out a range of contradictory functions. Similarly, here we describe the effects of the HpARI family and find that HpARI1 and HpARI2 bind HS and suppress IL-33 responses, while HpARI3 does not bind HS, and amplifies IL-33 responses. As the Hp-TGM proteins are targeted to different cellular targets through co-receptor interactions, we propose that the activity of the HpARIs is controlled by timing of release, in vivo half-life, and localisation within the host, controlled by interactions with the extracellular matrix.

We show here that HpARI1 and HpARI2 have a positively charged patch in their CCP1 domain. In the case of HpARI2, we show that this charged patch lengthens the effective half-life of the protein in vivo via binding to DNA and, more relevantly, HS. We propose that this binding to the extracellular matrix and cell surface proteoglycans prevents diffusion of HpARI1 and HpARI2, keeps the local concentration high, and prevents loss of protein to the circulation. This makes HpARI2 a particularly effective IL-33 suppressor when compared to a systemically administered monoclonal antibody which must reach an effective concentration throughout the circulation of the host to allow blockade at the site of IL-33 release. Some cytokines and chemokines (*Ridley et al., 2023*) bind to the extracellular matrix to ensure their effects are constrained to the local milieu. This is critical for their activity: it was recently shown that abrogating the HS-binding activity of IFN-γ results in fatal systemic (rather than local) inflammation after viral infection (*Kemna et al., 2023*). This approach of retaining proteins on the extracellular matrix is being developed for increasing the efficacy of protein-based therapeutics, while decreasing toxicity at off-target sites (*Alshoubaki et al., 2023*; *Martino et al., 2014*). This was exemplified in a study showing that heparin binding was able to prolong the blocking of VEGF signalling by recombinant VEGFR1 in therapeutic use (*Xin et al., 2021*). It appears that this same technique has been evolved by parasitic helminths to localise immunomodulatory proteins. HS binding

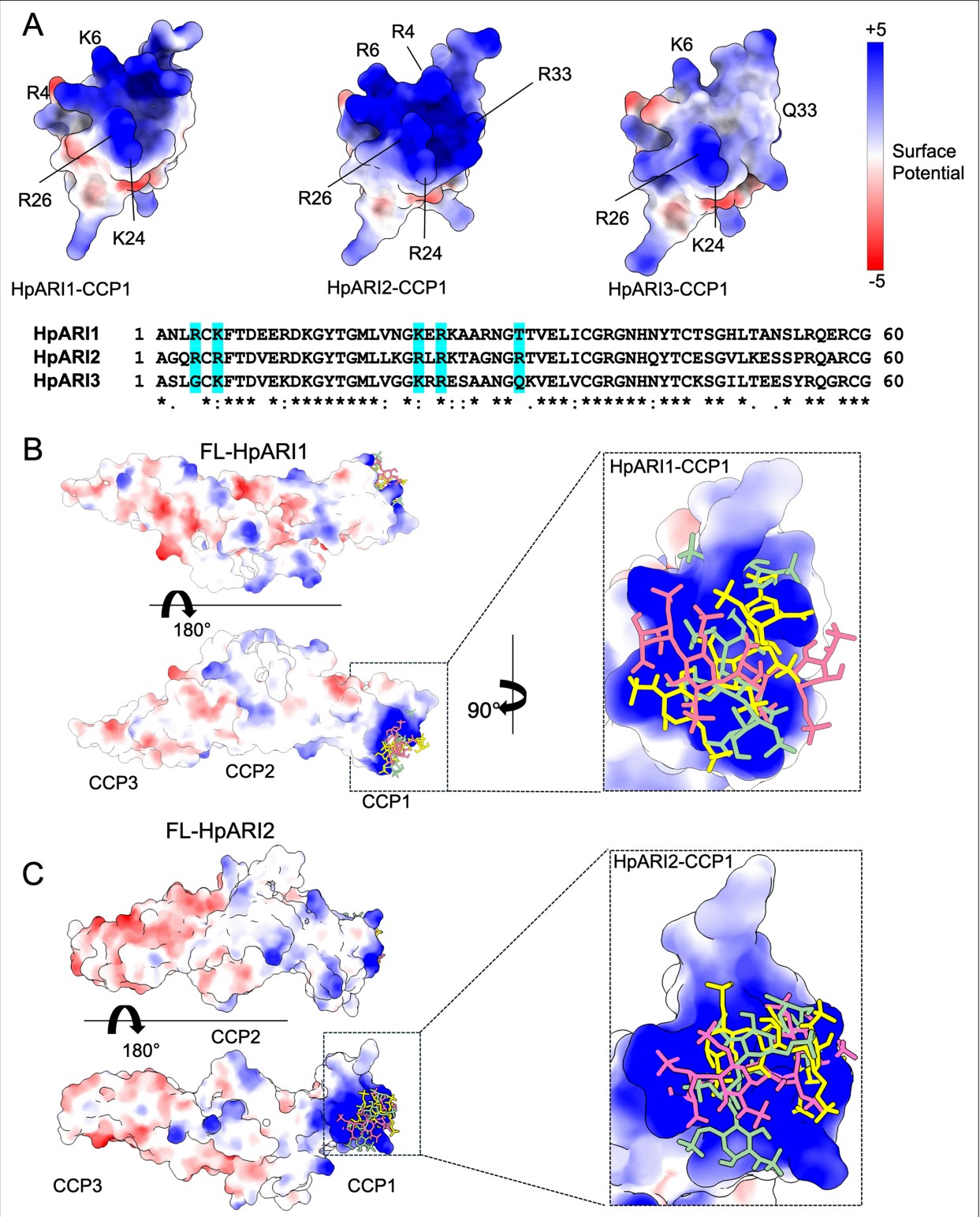

**Figure 5.** Molecular modelling of HpARI2 interaction of heparin oligosaccharide. (**A**) The top panel shows electrostatic surface rendering of AlphaFold models of the CCP1 domains of HpARI1-3. Blue and red surfaces indicate positive and negative surfaces, respectively. The lower panel shows an amino acid sequence alignment of HpARI family proteins with residues contributing to electropositive patch highlighted in cyan. (**B, C**) Electrostatic surface representation of an AlphaFold model of heparin tetrasaccharide docked on full-length HpARI1 (**B**) and HpARI2 (**C**). Three different docking solutions are shown. The right-hand panels show the spread of these models with heparins shown as coloured sticks on a surface representation of the CCP1 domain.

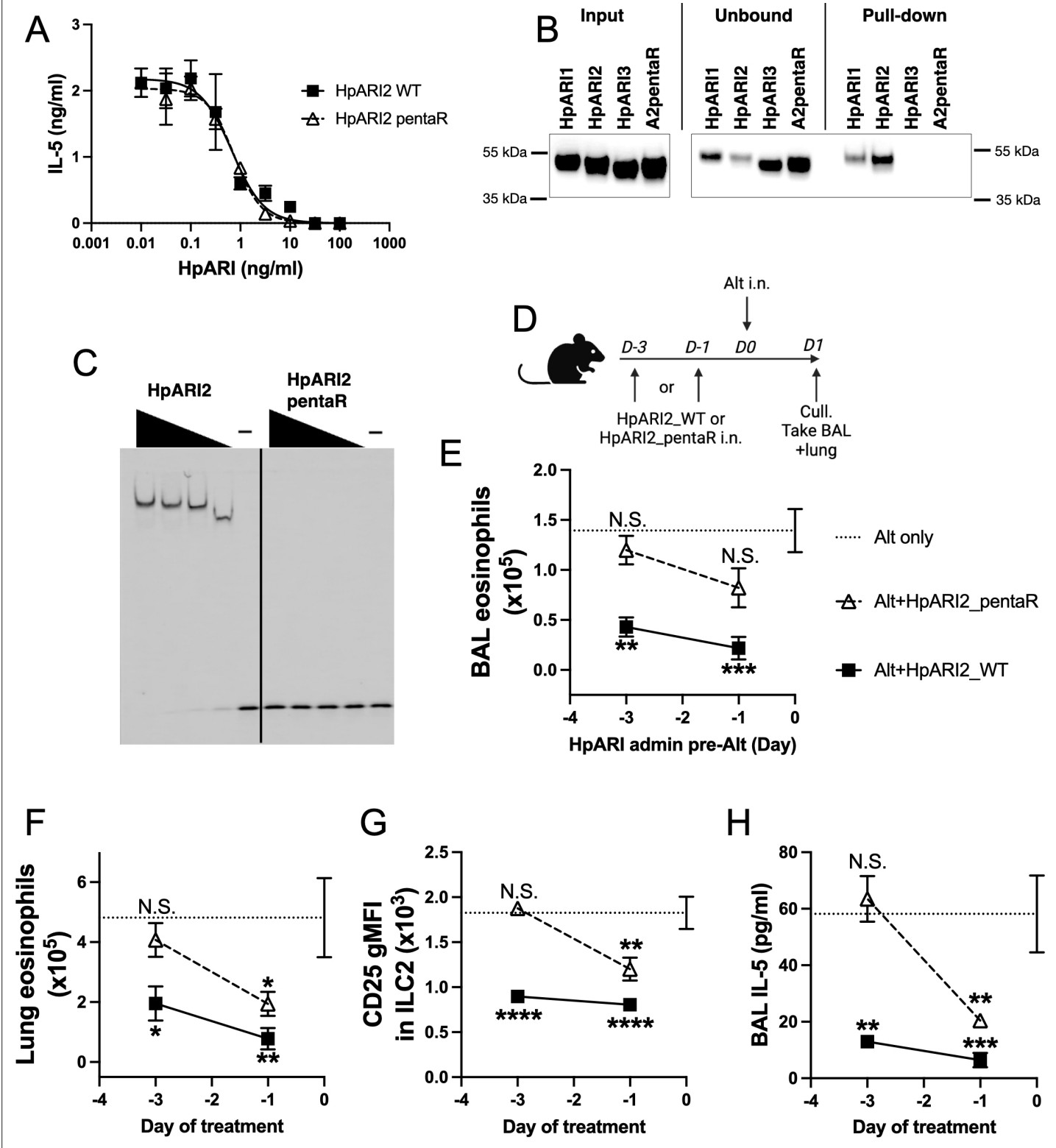

**Figure 6.** HpARI2 pentaR mutant effectively blocks IL-33 responses in vitro, but has a short half-life in vivo. (**A**) IL-5 production from bone marrow cells in response to IL-2, IL-7, and freeze-thawed CMT-64 supernatants, in the presence of a range of concentrations of HpARI2_WT or HpARI2_pentaR. Data pooled from three biological replicates. (**B**) Coomassie gel of HpARI1, HpARI2, HpARI3, or HpARI2-pentaR (A2pentaR) pull-down using heparin-coated beads. Input, unbound (i.e. supernatant from beads), and pull-down elution shown. Representative of three repeats. (**C**) Electromobility shift assay (EMSA) of HpARI2 or HpARI2_pentaR. Representative of two repeats. (**D**) Experimental setup for C–F. HpARI2_WT or HpARI2_pentaR (10 μg

*Figure 6 continued on next page*

*Figure 6 continued*

of each) were administered intranasally 3 days or 1 day prior to Alternaria (Alt) allergen. Mice were culled 24 hr after Alternaria administration, and bronchoalveolar lavage (BAL) and lung tissue taken for analysis. (**E**) BAL eosinophil numbers (Siglecf⁺CD11⁻CD45⁺) from mice treated as shown in D. (**F**) Eosinophil (SiglecfʰⁱCD11⁻CD45⁺) numbers in lung tissue from mice treated as shown in D. (**G**) CD25 geometric mean fluorescence intensity on lung ILC2 (ICOS⁺Lin⁻CD45⁺) from mice treated as shown in D. (**H**) BAL IL-5 levels (enzyme linked immunosorbent assay [ELISA]) from mice treated as shown in D. Data in E–H from a single experiment, for a total of four biological replicates per timepoint. Error bar shows SEM. NS = not significant, *=p<0.05 **=p<0.01, ***=p<0.001, ****=p<0.0001. Analysed by one-way analysis of variance (ANOVA) with Dunnett's post-test, comparing each condition to Alternaria-only control.

The online version of this article includes the following source data for figure 6:

**Source data 1.** Original Coomassie and electromobility shift assay (EMSA) gel images from *Figure 6B and C*.

**Source data 2.** Annotated Coomassie and electromobility shift assay (EMSA) gel images from *Figure 6B and C*.

has been demonstrated for several other CCP domain proteins, including the viral CCP protein Kaposi's sarcoma-associated herpesvirus complement control protein (KCP) (*Mark et al., 2006*), and the mammalian proteins Factor H (*Schmidt et al., 2008*) and C4BP (*Spijkers et al., 2008*; *Trouw et al., 2005*). In C4BP, the protein presents a basic patch which is able to bind both DNA and HS, similarly to the data we show here for HpARI1 and HpARI2.

While HpARI1 binds to DNA and HS, it has lower affinity for these ligands compared to HpARI2, and in the DNA-binding assays only bound to 50% of the total DNA. It is unclear what the purpose is of this reduced binding affinity, but it could be that HpARI1 remains partially in solution, diffusing further from the site of deposition than HpARI2, and so mediating its effects at a greater distance from the site of release.

In this context, it is interesting that HpARI3 does not bind HS, while having the opposite effect on IL-33 as compared to HpARI2. The effects of IL-33 are highly contingent on the context of IL-33 release, and the cell type and tissue that it is acting on. In parasitic infection, IL-33 released by epithelial cells in the intestine induces a type 2 immune response which results in parasite ejection. In the same helminth infection model, IL-33 released by myeloid cells induced regulatory T cell expansion, suppressing effector immune responses and resulting in increased parasite burdens (*Hung et al., 2020*). Therefore, at different sites and in different cells, IL-33 can have directly opposing effects.

In the context of the findings outlined here, we propose the following model for the HpARI family's effects: HpARI2 binds strongly to the extracellular matrix, and is retained in the local milieu of the parasite where it blocks IL-33 released in response to infection. HpARI1 binds less strongly to HS, and therefore may diffuse slightly further from the site of deposition, to block more distant IL-33 responses. Finally, HpARI3 lacks HS binding and so diffuses freely to amplify IL-33 responses at distal sites, which we hypothesise could amplify IL-33-dependent regulatory responses. To investigate these hypotheses, transgenic parasites deficient for individual HpARI proteins would be useful, however this has proved intractable for parasitic nematodes (*Quinzo et al., 2022*), and has not been achieved in *H. polygyrus bakeri* to date.

The findings outlined here may be applicable beyond *H. polygyrus bakeri*, as it appears likely that other parasitic helminths have used extracellular matrix binding as a technique to retain immunoregulators within the local milieu. Indeed, *Trichuris muris*, another intestinal nematode, secretes the immunoregulatory protein p43. This protein binds and blocks IL-13, but also has HS-binding activity which has been proposed to tether p43 to the extracellular matrix, increasing survival of the protein at the site of infection (*Bancroft et al., 2019*). *H. polygyrus bakeri* has developed a series of immunomodulatory protein families, including 10 Hp-TGMs and 2 HpBARIs, therefore it will be informative to investigate whether members of these families also bind to the extracellular matrix, and whether this alters their in vivo localisation and/or half-life.

In conclusion, we describe a novel HS-binding ability of members of the HpARI family and show that this binding increases their in vivo half-life and activity at the site of deposition.

## Materials and methods
### Expression and purification of HpARI proteins and mutants

Proteins were expressed and purified as described previously (*Chauché et al., 2020*; *Osbourn et al., 2017*; *Vacca et al., 2020*). Briefly, inserts encoding the gene of interest (HpARI1,2,3, HpARI3:2,

HpARI2:3, and HpARI_pentaR mutants) were cloned into the pSecTAG2A mammalian expression vector using AscI and NotI-HF restriction enzyme sites, to create constructs encoding these proteins with a C-terminal c-myc epitope tag and 6X-His-tag. These constructs were each transfected into Expi293F cells using the ExpiFectamine Transfection Kit (Thermo Fisher Scientific). Cell supernatants were harvested 96 hr post-transfection, and the expressed recombinant proteins were then captured from the filtered supernatants using Ni-NTA chromatography, then dialysed to PBS and filter sterilised. The HEK-Blue TLR4 reporter assay (Invivogen) was used to assess endotoxin contamination, and levels of all purified proteins were below the detection limit of the assay (<0.01 EU LPS per μg protein).

## Animals

Female BALB/cAnNCrl and C57BL/6JCrl mice were purchased from Charles River, UK, and were used at 6–12 weeks of age. Experiments were cage blocked: each cage contained one member of each group in the experiment, thus controlling for cage effects. Mouse accommodation and procedures were performed under UK Home Office licenses (PP9520011) with institutional oversight performed by qualified veterinarians and with approval of the local Welfare and Ethical Use of Animals Committee (WEC). Animal research carried out in accordance with ARRIVE guidelines. Power calculations were carried out prior to experiments to determine appropriate samples sizes for an 80% power to achieve statistical significance (p<0.05) at the effect size expected.

## In vivo Alternaria model challenge

BALB/c mice were intranasally administered with 50 μg Alternaria allergen (Greer XPM1D3A25) and 10 μg HpARI wild-type or mutant proteins suspended in PBS, carried out under isoflurane anaesthesia. For most experiments, HpARI proteins and Alternaria allergen were co-administered, but in indicated experiments HpARI proteins were administered 1, 3, or 7 days prior to Alternaria. For IL-33 measurements in BAL, mice were culled 15 min later, and BAL was collected (one lavage with 0.5 ml ice-cold PBS for cytokine measurements only). IL-33 levels were quantified in undiluted BAL fluid by ELISA following the manufacturer's instructions (R&D Systems) and western blot using anti-IL-33 antibody (R&D Systems). For later immunological response analysis, mice were culled 24 hr after Alternaria+HpARI administration, and BAL was collected (four lavages with 0.5 ml ice-cold PBS for cellular and cytokine measurements). Lungs were taken for single-cell preparation and flow cytometry, as previously described (*Colomb et al., 2024*; *Osbourn et al., 2017*). IL-5 levels were quantified in undiluted BAL fluid by ELISA following the manufacturer's instructions (Invitrogen).

## Single-cell preparations and flow cytometry

Single-cell preparations of lung tissue were prepared as described previously (*Osbourn et al., 2017*). Lungs were digested by shaking for 35 min at 37°C in 2 U/ml Liberase TL (Sigma) and 80 U/ml DNAseI (Thermo Fisher Scientific) in PBS. Red blood cells in lung and BAL single-cell preparations were lysed using ACK buffer, then total viable cell numbers (using trypan blue staining) calculated from haemacytometer cell counts. Cells were stained for viability with Zombie UV Fixable Viability Kit (BioLegend) according to the manufacturer's instructions, blocked with anti-mouse CD16/CD32 (clone 93; BioLogend) and stained for either ILC2 or eosinophil markers. ILC2 markers: CD45 (AF700, clone 30-F11, or APC-Cy7, I3/2.3; BioLegend), CD90.2 (AF700, clone 30-H12, BioLegend), ICOS (PE, clone C398.4A; BioLegend), lineage markers (all on FITC: CD3ε, clone 145-2C11; CD5, clone 53-7.3; CD11b, clone M1/70.15; CD19, clone 6D5; CD49b, clone DX5; BioLegend), CD25 (PerCP, clone PC61; BioLegend), and ST2 (APC, clone RMST2-2; Invitrogen). Eosinophil markers: CD45 (AF700, clone 30-F11, or APC-Cy7, I3/2.3; BioLegend), SiglecF (PE, clone REA798, Miltenyi), CD11c (AF647, clone N418, BioLegend). Cells were then acquired on an LSR Fortessa (BD) and analysed on FlowJo v10.9 (BD). Gating strategies and representative flow cytometry plots are shown in *Figure 3—figure supplements 1–3*. BAL eosinophils were gated on SiglecF⁺CD11c⁻CD45⁺, lung recruited inflammatory eosinophils (*Mesnil et al., 2016*) were gated on SiglecF$^{hi}$CD11c⁻CD45⁺. ILC2s were gated on ICOS⁺Lineage⁻CD45⁺ or ICOS⁺CD90.2⁺Lineage⁻CD45⁺ as indicated. Absolute cell numbers were calculated by gating on the population of interest and multiplying the percentage of this population within the live cell gate by the haemacytometer total viable cell count.

## Electrophoretic mobility shift assays

IR700-conjugated oligonucleotides were manufactured by Integrated DNA Technologies (IR700-5'-AACTTTGCCATTGTGGAAGG-3', 5'-CCTTCCACAATGCCAAAGTT-3'). 50 fmol of oligonucleotides were incubated in binding buffer (10 mM Tris, 50 mM KCl, 1 mM DTT, 2.5% glycerol, 5 mM $MgCl_2$, 0.05% NP40, pH 7.5) with the indicated concentrations of HpARI proteins for 30 min at room temperature. Samples were then run on a 5% gel (Acrylamide/Bisacrylamide 37.5:1 Protogel, National Diagnostics) in 0.5% TBE buffer. Images were then acquired on a LiCor Odyssey Fc using the IR 700 channel.

## Heparin-agarose pull-down assays

Heparin-agarose beads (#H6508, Sigma) were washed twice with PBS 0.02% Tween 20 and incubated with 10 µg of HpARI proteins in a final volume of 100 µl of PBS 0.02%-Tween 20. After 30 min of incubation at room temperature, the beads were spun down and the supernatant collected. After three washes with PBS 0.02% Tween 20, the bound fraction was eluted in 1X Loading Sample Buffer (Thermo Fisher Scientific) containing 5% 2-mercaptoethanol, heated to 70°C for 5 min and ran on a 4–12% NuPAGE precast gel (Thermo Fisher Scientific). Proteins were then detected using InstantBlue Coomassie Stain (Abcam) following the manufacturer's instructions.

## CMT-64 cells culture and treatment

CMT-64 cells (ECACC 10032301) were maintained in complete RPMI (RPMI 1640 medium containing 10% fetal bovine serum, 2 mM L-glutamine, 100 U/ml penicillin, and 100 µg/ml streptomycin [Thermo Fisher Scientific]) at 37°C, 5% $CO_2$; 96-well plates were seeded at $5 \times 10^4$ cells/well. Cells were grown to 100% confluency, washed, and incubated in complete RPMI containing HpARI proteins at the concentrations indicated. Cells were snap-frozen on dry ice for at least 1 hr, then thawed and incubated at 37°C for 2 hr, prior to the collection of supernatants and application to bone marrow cell cultures, as described previously (*Colomb et al., 2024*).

## Bone marrow assay and ELISA

Bone marrow single-cell suspensions were prepared from C57BL/6 mice by flushing tibias and femurs with RPMI 1640 medium using a 21 g needle. Cells were resuspended in ACK lysis buffer (Gibco) for 5 min at room temperature, prior to resuspension in complete RPMI (with 10% FCS, 1% penicillin/streptomycin, 1% L-glutamine; Gibco) and passing through a 70 µm cell strainer. Cells were cultured in round-bottom 96-well plates in a final 200 µl volume, containing $0.5 \times 10^6$ cells/well. IL-2 and IL-7 (BioLegend) were added at a 10 ng/ml final concentration. CMT freeze-thaw supernatant prepared as described earlier was added at 50 µl/well. Cells were then cultured at 37°C, 5% $CO_2$, for 5 days, prior to supernatant collection. IL-5 concentration was assessed using mouse-uncoated IL-5 ELISA kits (Invitrogen) following the manufacturer's instructions.

## Gel filtration

50 µg of HpARI proteins were incubated with 50 µg of HS (HS sodium salt from bovine kidney, Sigma) in PBS for 30 min at room temperature. The mix was then run through a Superdex 200 Increase 10/300 GL gel filtration column (Cytiva) and detected by monitoring of UV A280 nm absorbance.

## Cell-binding assays

HpARI proteins tetramers were prepared by mixing HpARI biotinylated proteins with PE-conjugated streptavidin (BioLegend) at a 4:1 molar ratio for 30 min at room temperature. BALB/c mice lungs were taken for single-cell preparation and flow cytometry, as previously described (*Osbourn et al., 2017*). Cells were washed twice in PBS and stained with Zombie UV Fixable Viability Kit (BioLegend) according to the manufacturer's instructions. Cells were then incubated for 30 min with anti-mouse CD16/CD32 (clone 93; BioLegend) for Fc receptors blocking and then simultaneously incubated with HpARI proteins tetramers and stained for CD45 (APC-Cy7, clone 30-F11; BioLegend). Cells were then acquired on an LSR Fortessa (BD) and analysed on FlowJo v10.9 (BD).

## Structural modelling

Full-length models of HpARI1-3 were built using AlphaFold with crystal structure of HpARI2_CCP2/3 (PDB ID = 8Q5R) as a template (*Tunyasuvunakool et al., 2021*). Top-ranking, relaxed AlphaFold

models were selected for further analysis. Electrostatic potential maps of the HpARI1-3 proteins were generated from the AlphaFold model and visualised using ChimeraX (*Pettersen et al., 2021*). The ClusPro server (https://cluspro.org) was used to predict the location of the heparin-binding site on HpARI2. Representative heparin-HpARI2 complex was selected based on scoring criteria described previously (*Kozakov et al., 2017*).

### Isothermal calorimetry

HpARI1, HpARI2, and HpARI3 were first size exclusion chromatography (SEC) purified in a buffer containing 10 mM HEPES pH 7.2 and 100 mM NaCl to remove misfolded protein aggregates. Proteins were then concentrated to 50 µM for isothermal calorimetry experiments (ITC). All ITC experiments were performed on Microcal PEAQ-ITC (Malvern) and Microcal PEAQ-ITC analysis software was used for data analysis. Direct titration of proteins was done individually against HS tetrasaccharide (HS4) (Iduron, UK) reconstituted in SEC buffer. Typical titration was carried over 20 injections with an initial injection of 0.4 µl followed by 19 injections of 2 µl of 2 mM HS tetrasaccharide into protein solution inside the cell, stirring at 750 rpm at room temperature. All titrations were done twice.

### Statistics

Statistical data were analysed using GraphPad Prism v10.0.0. When comparing independent groups, one-way analysis of variance (ANOVA) with Dunnett's post-test was used. When comparing groups over a timecourse, two-way ANOVA with Dunnett's post-test was used. Standard error of mean is used throughout. ****=$p<0.0001$, ***=$p<0.001$, **=$p<0.01$, *=$p<0.05$, ns = not significant ($p>0.05$).

### Materials availability statement

All materials created in this manuscript may be accessed by contacting the corresponding author, under appropriate material transfer agreements.

## Acknowledgements

This work was funded by awards to HJM from LONGFONDS | Accelerate as part of the AWWA project, the Medical Research Council (MR/S000593/1), and Wellcome (221914/Z/20/Z).

## Additional information

### Funding

| Funder | Grant reference number | Author |
|---|---|---|
| Wellcome Trust | 10.35802/221914 | Henry J McSorley |
| Longfonds | Accelerate as part of the AWWA project | Henry J McSorley |
| Medical Research Council | MR/S000593/1 | Henry J McSorley |

The funders had no role in study design, data collection and interpretation, or the decision to submit the work for publication. For the purpose of Open Access, the authors have applied a CC BY public copyright license to any Author Accepted Manuscript version arising from this submission.

### Author contributions

Florent Colomb, Abhishek Jamwal, Conceptualization, Investigation, Methodology, Writing - original draft, Writing - review and editing; Adefunke Ogunkanbi, Tania Frangova, Alice R Savage, Sarah Kelly, Investigation, Methodology; Gavin J Wright, Conceptualization, Methodology; Matthew K Higgins, Conceptualization, Supervision; Henry J McSorley, Conceptualization, Supervision, Funding acquisition, Investigation, Methodology, Writing - original draft, Project administration, Writing - review and editing

### Author ORCIDs

Gavin J Wright ⓘ https://orcid.org/0000-0003-0537-0863

Matthew K Higgins (iD) https://orcid.org/0000-0002-2870-1955
Henry J McSorley (iD) https://orcid.org/0000-0003-1300-7407

### Ethics

Mouse accommodation and procedures were performed under UK Home Office license (project license PP9520011) with institutional oversight performed by qualified veterinarians. Experiments were reviewed by the local Animal Welfare Ethical Review Body (AWERB) and were preformed in accordance with Laboratory Animal Science Association (LASA) guiding principals.

Reviewer #1 (Public Review): https://doi.org/10.7554/eLife.99000.3.sa1
Reviewer #2 (Public Review): https://doi.org/10.7554/eLife.99000.3.sa2
Author response https://doi.org/10.7554/eLife.99000.3.sa3

---

## Additional files

### Supplementary files

• MDAR checklist

### Data availability

No new datasets were generated for this study; source data files have been provided for *Figures 1, 2, 4 and 6*.

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
