## [Editor Report · eLife Assessment]

This **important** study uses in vitro and in vivo methods to identify HpARI proteins from H. polygyrus as modulators of the host immune system. The data from comprehensive approaches for investigating differential roles of HpARI proteins are **convincing**. This paper is relevant to those who investigate host-pathogen interactions at the systems and molecular levels.

---

## [Referee Report · Reviewer #1 (Public Review)]

Colomb et al have further explored the mechanisms of action of a family of three immunodulatory proteins produced by the murine gastrointestinal nematode parasite Heligmosomoides polygyrus bakeri. The family of HpARI proteins binds to the alarmin interleukin 33 and depending on family members, exhibits differential activities, either suppressive or enhancing. The present work extends previous studies by this group showing the binding of DNA by members of this family through a complement control protein (CCP1) domain. Moreover, they identify two members of the family that bind via this domain in a non-specific manner to the extracellular matrix molecule heparan sulphate through a basic charged patch in CCP1. The authors thus propose that binding to DNA or heparan sulphate extends the suppressive action of these two parasite molecules, whereas the third family member does not bind and consequently has a shorter half-life and may function via diffusion.

---

## [Referee Report · Reviewer #2 (Public Review)]

Colomb et al. investigated here the heparin-binding activity of the HpARI family proteins from H. polygyrus. HpARIs bind to IL-33, a pleiotropic cytokine, and modulate its activities. HpARI1/2 has suppressive functions, while HpARI3 can enhance the interaction between IL-33 and its receptor. This study builds upon their previous observation that HpARI2 binds DNA via its CCP1 domain. Here, the authors tested the CCP1 domain of HpARIs in binding heparan sulfate, an important component of the extracellular matrix, and found that 1/2 bind heparan, but 3 cannot, which is related to their half-lives in vivo.

---

## [Author Response]

The following is the authors’ response to the original reviews.

**Reviewer #1 (Public Review):**
Summary:Colomb et al have further explored the mechanisms of action of a family of three immunodulatory proteins produced by the murine gastrointestinal nematode parasite Heligmosomoides polygyrus bakeri. The family of HpARI proteins binds to the alarmin interleukin 33 and depending on family members, exhibits differential activities, either suppressive or enhancing. The present work extends previous studies by this group showing the binding of DNA by members of this family through a complement control protein (CCP1) domain. Moreover, they identify two members of the family that bind via this domain in a non-specific manner to the extracellular matrix molecule heparan sulphate through a basic charged patch in CCP1. The authors thus propose that binding to DNA or heparan sulphate extends the suppressive action of these two parasite molecules, whereas the third family member does not bind and consequently has a shorter half-life and may function via diffusion.Strengths:A strength of the work is the multifaceted approach to examining and testing their hypotheses, using a well-established and well-defined family of immunomodulatory molecules using multiple approaches including an in vivo setting.Weaknesses:There are a few weaknesses of the approach. Perhaps some discussion and speculation as to how these three family members might operate in concert during Heligmosomoides polygyrus bakeri infection would help place the biology of these molecules in context for the reader, e.g. when and where they are produced.

We agree that the roles of these proteins during infection requires further study and is not fully elucidated in infection here. We have added further discussion to the manuscript on their potential roles during infection (track changes manuscript, lines 277 – 283).

**Reviewer #2 (Public Review):**
Summary:Colomb et al. investigated here the heparin-binding activity of the HpARI family proteins from H. polygyrus. HpARIs bind to IL-33, a pleiotropic cytokine, and modulate its activities. HpARI1/2 has suppressive functions, while HpARI3 can enhance the interaction between IL-33 and its receptor. This study builds upon their previous observation that HpARI2 binds DNA via its CCP1 domain. Here, the authors tested the CCP1 domain of HpARIs in binding heparan sulfate, an important component of the extracellular matrix, and found that 1/2 bind heparan, but 3 cannot, which is related to their half-lives in vivo.Strengths:The authors use a comprehensive multidisciplinary approach to assess the binding and their effects in vivo, coupled with molecular modeling.Weaknesses:(1) Figure 1C should include Western.

We apologise for this oversight, and now include an uncropped western blot image as a Figure 1, Figure Supplement 1.

(2) Figure 1E: Why does HpARI1 stop binding DNA at 50%?

It is currently unclear why HpARI1 does not bind to all DNA in the EMSA assay, however this was our repeated finding. With our revised findings we can now state definitively that HpARI1 has a lower affinity for HS compared to HpARI2, and in each of our assays (EMSA (Fig 1D-E), size exclusion chromatography (Fig 4A), HS-bead pull-down (Fig 4B), lung cell surface binding (Fig 4C) and ITC (Fig 4D)) HpARI1 always shows a weaker response compared to HpARI2. We hypothesise that HpARI1 binds more weakly to DNA/HS to allow it to diffuse further from the site of deposition, but we have yet to demonstrate this during infection. We add further discussion of this point (track changes manuscript, lines 262 – 266).

(3) ITC binding experiment with HpARI1? Also, the ITC results from HpARI2 do not seem to saturate, thus it is difficult to really determine the affinity.

We have now included HpARI1-HS ITC, and re-ran the HpARI2 experiment to saturation (Fig 4D-E).

(4) It would be helpful to add docking results from HpARI1.

We have now included HpARI1-HS docking, in Figure 5B.

(5) Some conclusions are speculative and need to remain in the Discussion. e.g.: a That HpARI3 may be able to diffuse farther

We have rewritten these points to remove the speculation on localisation from the abstract (lines 18-19) and introduction (line 78).

b) That DNA/HS may trap HpARI1/2 at the infection site.

Likewise, these points have been rewritten in the abstract and introduction as above, and we have made it clearer that this is a model that we are proposing in the discussion (line 277-283).

**Reviewer #1 (Recommendations For The Authors):**
The paper is well-written and the data well-presented. I have one small comment that the authors may like to consider. In the discussion, second paragraph, line 17, perhaps, "evolved" rather than "developed".

Thank you for this suggestion, we have made this change (line 248).